# Attosecond delays between dissociative and non-dissociative ionization of polyatomic molecules

Xiaochun Gong [1,2] ✉, Étienne Plésiat [3], Alicia Palacios [3,4], Saijoscha Heck[1], Fernando Martín [3,5,6] ✉ & Hans Jakob Wörner [1] ✉

The interplay between electronic and nuclear motions in molecules is a central concept in molecular science. To what extent it influences attosecond photoionization delays is an important, still unresolved question. Here, we apply attosecond electron-ion coincidence spectroscopy and advanced calculations that include both electronic and nuclear motions to study the photoionization dynamics of $CH_4$ and $CD_4$ molecules. These molecules are known to feature some of the fastest nuclear dynamics following photoionization. Remarkably, we find no measurable delay between the photoionization of $CH_4$ and $CD_4$, neither experimentally nor theoretically. However, we measure and calculate delays of up to 20 as between the dissociative and non-dissociative photoionization of the highest-occupied molecular orbitals of both molecules. Experiment and theory are in quantitative agreement. These results show that, in the absence of resonances, even the fastest nuclear motion does not substantially influence photoionization delays, but identify a previously unknown signature of nuclear motion in dissociative-ionization channels. These findings have important consequences for the design and interpretation of attosecond chronoscopy in molecules, clusters, and liquids.

The way electrons and nuclei share their energy in a molecule is a central concept in molecular physics and chemistry, since it is at the root of many fundamental properties and dynamical processes in matter. Prominent manifestations of this effect include, e.g., conical intersections[1], the Jahn-Teller effect (JTE)[2,3], or electron-phonon coupling in solids[4]. It is well-established that the inclusion of both electronic and nuclear motions is a prerequisite for even a qualitatively correct description of excited-state dynamics in molecules[5]. This has also been suggested to be the case in molecular photoionization, in particular in the presence of shape resonances[6], where the coupling between the fast electronic and the slow nuclear motions can be considerably enhanced owing to the extended trapping time of the

photoelectron before its escape[7–12]. Whereas the profound impact of electron-nuclear couplings on excited-state dynamics is a direct consequence of the relatively long time scales at play[13–17], its influence on attosecond photoionization dynamics remains unknown[18,19].

We introduce two new ideas that allow us to directly access and quantify the effect of nuclear motion on molecular photoionization delays. First, we perform a direct comparison of delays associated with dissociative and non-dissociative ionization to the same final electronic state of the molecular cation. This eliminates the electronic effects caused by ionization from different orbitals and isolates the effect of nuclear motion in the measurement. Second, we also introduce the direct comparison of delays from different isotopologues ($CH_4$ vs.

[1]Laboratorium für Physikalische Chemie, ETH Zürich, 8093 Zürich, Switzerland. [2]State Key Laboratory of Precision Spectroscopy, East China Normal University, Shanghai 200241, China. [3]Departamento de Química, Módulo 13, Universidad Autónoma de Madrid, 28049 Madrid, Spain. [4]Institute of Advanced Research in Chemical Sciences (IAdChem), Universidad Autónoma de Madrid, 28049 Madrid, Spain. [5]Instituto Madrileño de Estudios Avanzados en Nanociencia (IMDEA Nano), Cantoblanco, 28049 Madrid, Spain. [6]Condensed Matter Physics Center (IFIMAC), Universidad Autónoma de Madrid, 28049 Madrid, Spain. ✉e-mail: xcgong@lps.ecnu.edu.cn; fernando.martin@uam.es; hwoerner@ethz.ch

$CD_4$) to quantify the effect of the nuclear mass on the photoionization delays. Since the photoelectron leaves the parent ion within a few tens of attoseconds at typical photon energies in the extreme ultraviolet (XUV), only the fastest nuclear motions can be expected to leave an observable imprint on the photoionization delays. To maximize these effects, we chose a molecule that features some of the fastest nuclear dynamics following ionization, i.e., $CH_4$. Since photoionization from the highest-occupied molecular orbital (HOMO) leaves $CH_4^+$ in a triply degenerate electronic state ($^2T_2$ in the $T_d$ point group), it is subject to a Jahn-Teller effect, which distorts the molecule from the tetrahedral geometry of the neutral ground state to a $C_{2v}$-symmetric equilibrium geometry[20,21]. This distortion is accompanied by an unusually large energetic stabilization of ~1.5 eV[22]. Since this stabilization arises from the displacement of hydrogen (or deuterium) atoms only, it drives some of the fastest ionization-induced motions in any known system with a nuclear auto-correlation function decaying to zero within 2 fs[23,24]. An indication of this unusually fast nuclear motion has been obtained in experiments using high-harmonic spectroscopy[25]. The strong JTE in $CH_4^+$ is also responsible for the direct dissociative ionization of the molecule following HOMO ionization. This offers the possibility to study delays between dissociative and non-dissociative ionization channels arising from ionization to the same ($^2T_2$) final state of the cation.

Here, we use the reconstruction of attosecond beating by interference of two-photon transitions (RABBIT) technique[26,27] in conjunction with cold-target recoil ion-momentum spectroscopy (COLTRIMS)[28,29]. Details on the experimental setup are given in the Methods Section and in refs. [11,30]. We simultaneously measure photoionization delays between photoelectron wave packets that leave behind intact $CH_4^+/CD_4^+$ cations, or the dissociating fragments $CH_3^+$ +H/$CD_3^+$ +D. This allows us to quantify the effects of nuclear motion in both non-dissociative and dissociative ionization events that always involve the removal of an electron from the HOMO. The measurements are interpreted through comparison with calculated RABBIT spectra, in which the ionization process is described by explicitly including the electronic continuum of the molecule and the continuum-continuum transitions induced by the laser pulses, but also, and more importantly, by incorporating the effect of nuclear motion along the $CH_3^+$ +H/$CD_3^+$ +D dissociation pathways, thus accounting for variations of the ionization probabilities and photoionization delays during that motion. These calculations quantitatively agree with the experimental results. Whereas we find delays of up to 20 attoseconds between the dissociative and non-dissociative ionization of both molecules, we find no measurable delays between $CH_4$ and $CD_4$. These results indicate a non-negligible, previously unknown phase shift between the electronic continua associated with dissociative and non-dissociative channels. The observed effects are consistent with photoionization delays that increase with the spatial delocalization of the orbital from which ionization takes place.

## Results and Discussion

As shown in Fig. 1a, the removal of an electron from the $1t_2$ HOMO of $CH_4$ leaves $CH_4^+$ in a triply degenerate $^2T_2$ state subject to a JTE that lifts its electronic degeneracy. As a consequence, the Franck-Condon region corresponds to a three-fold conical intersection between the lowest-lying electronic states of the cation. The photoelectron band consists of a broad, dual-hump structure extending over more than 1 eV. A cut through the potential-energy surfaces along the C-H bond-dissociation coordinate is shown in Fig. 1b. Our experiments make use of an extreme-ultraviolet attosecond pulse train (XUV-APT) synthesized either from harmonics 11, 13, and 15, obtained from high-harmonic generation of an 800-nm (NIR) driving pulse in xenon, or from harmonics 11 through 25, generated in argon. The APT and NIR pulses are overlapped in the interaction region of a COLTRIMS spectrometer, which detects the photoelectrons and photoions in

coincidence, as shown in Fig. 1c. The corresponding data obtained from the action of the XUV-APT alone are shown in Fig. 1d, e. The total photoelectron spectrum shown as the grey line in Fig. 1d displays the expected shape consisting of three replicas of the HOMO photoelectron band. The photoelectron spectra detected in coincidence with $CH_4^+$ and $CH_3^+$ as blue/orange-filled spectra nicely show that the upper half of the photoelectron band corresponds to non-dissociative ionization, whereas the lower part corresponds to dissociative ionization leading to $CH_3^+$ +H. Figure 1e illustrates the energy sharing between photoions and photoelectrons in terms of the kinetic energy of the molecular cation ($E_{mol}$, vertical axis) and the electron-kinetic energy ($E_e$, horizontal axis).

Figure 2a and b show the theoretically calculated and experimentally measured attosecond photoelectron spectrum in the non-dissociative channel $CH_4^+$ +$e^-$ obtained from an XUV-APT generated in argon, covering the photon-energy range from H11 (17.05 eV) to H25 (38.75 eV). Contributions from HOMO-1 ($2a_1$) are not observed because of the much smaller cross-section (25 times smaller than HOMO at 25 eV - see Supplementary Material (SM), Fig. S7) and can be excluded because HOMO-1 ionization leads to the formation of $CH_2^+$ fragments, which were not observed in our experiments.

These measurements all display the expected oscillations of the form $SB(\tau) = A_0\cos(2\omega_{NIR}\tau + \phi_0) + B_0$, which were analyzed by Fourier transformation along the delay axis (Fig. 2c), followed by analysis of the amplitude (orange) and phase (blue) of the Fourier transform at the $2\omega_{NIR}$ frequency. Details are given in the SM, Section 2. This approach has previously been shown to correctly account for the spectral overlap in RABBIT measurements[11,30–32]. The NIR intensity is estimated to be around 1 TW/cm$^2$, which is confirmed by the absence of Fourier frequencies higher than $2\omega_{NIR}$ in all experiments described herein. With the help of the COLTRIMS measurements on a mixture of $CH_4$ and $CD_4$, we were able to simultaneously record RABBIT data in coincidence with $CH_4^+$ (Fig. 2a–d), $CD_4^+$ (Fig. 2e–h), $CH_3^+$ +H (Fig. 2i–l) and $CD_3^+$ +D (Fig. 2m–p). In the case of the non-dissociative channels (panels a–h) the data is plotted as a function of the electron-kinetic energy. In the case of the dissociative channels (panels i–p) the data is plotted as a function of the kinetic energy sum $E_{sum} = E_e + E_{ion}$, where $E_e$ and $E_{ion}$ are the kinetic energies of the electron and $CH_3^+$, due to the electron-nuclear energy sharing (see SM, Fig. S1 for additional data).

Since the data presented in Fig. 2 were all acquired simultaneously, we can extract relative phases, hence relative photoionization delays between any pair of data, $\tau^{CH_3^+/H-CH_4^+} = \left(\phi_0^{CH_3^+/H} - \phi_0^{CH_4^+}\right)/(2\omega_{NIR})$, $\tau^{CH_4^+-CD_4^+} = \left(\phi_0^{CH_4^+} - \phi_0^{CD_4^+}\right)/(2\omega_{NIR})$, etc. Here, we first discuss the delays between dissociative and non-dissociative channels. Figure 3a and b show the experimentally measured time-delay difference $\tau^{CH_3^+/H-CH_4^+}$. Delays of up to 20 as were measured near the ionization threshold at the photon energy of 18.6 eV. These time delays decrease as a function of the photon energy. The calculated delays, extracted from fits of the sidebands appearing in the theoretical RABBIT spectra shown in Fig. 2a, agree very well with the experimentally measured results.

We now turn to the time delay difference between $CH_4$ and $CD_4$. Figure 4a and b show the time delay differences $\tau^{CD_4^+-CH_4^+}$ and $\tau^{CD_3^+/D-CH_3^+/H}$. In the non-dissociative ionization channels, the experimental results $\tau_{exp}^{CD_4^+-CH_4^+}$ show no measurable delays within the error bars, and the calculated time delays $\tau_{theo}^{CD_4^+-CH_4^+}$, extracted from the calculated RABBIT spectra, remain below 2 as over the entire investigated photon-energy range. The one-photon Wigner time-delay difference $\tau_{Wig}^{CD_4^+-CH_4^+}$ also agrees reasonably well with the experimental results and those of the complete theory, but it becomes negative at very low photon energies, whereas the full theory gives small positive delays, in agreement with the experimental data.

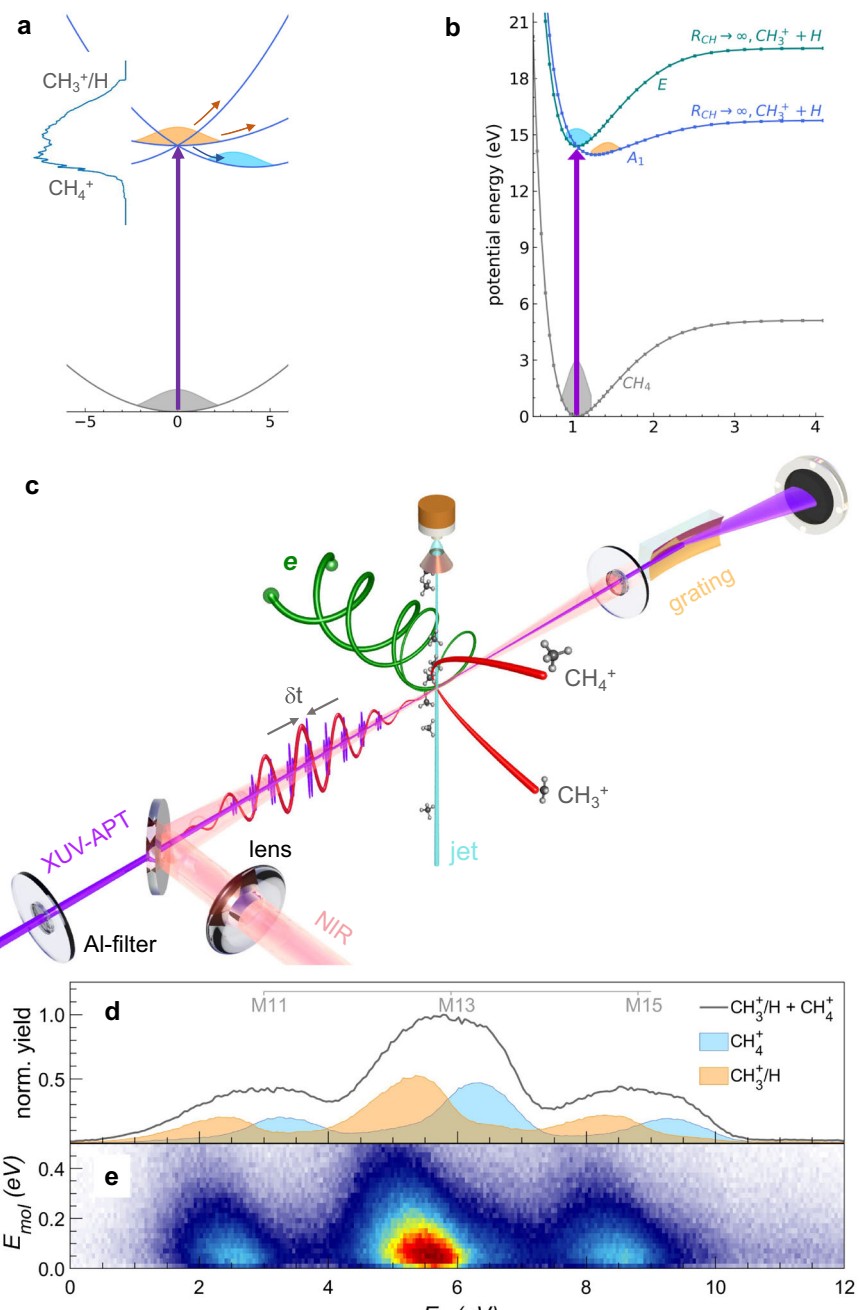

**Fig. 1 | Overview of dissociative and non-dissociative ionization of methane and its measurement by attosecond electron-ion-coincidence spectroscopy.**
**a** Sketch of the potential-energy surfaces of methane cation in the Franck-Condon region. The JTE causes the appearance of a three-fold conical intersection and a broad photoelectron band (taken from ref. 36) associated with HOMO ionization. **b** One-dimensional cut through the potential-energy surfaces along the C-H bond-dissociation coordinates obtained from CASSCF/MRCI calculations by relaxing all other nuclear-geometry parameters. **c** Schematic diagram of the attosecond coincidence interferometer. The phase-locked XUV-APT pump and NIR probe pulse serve as an attosecond clock to monitor the ultrafast electron dynamics at the instant of photoionization. **d** Measured photoelectron spectra following photoionization with an XUV-APT generated in xenon and containing harmonic orders 11, 13, and 15. **e** Energy sharing between photoion and photoelectron is represented in terms of their respective kinetic energies $E_e$ and $E_{mol}$.

In the dissociative channel, the theoretical results $\tau_{\text{theo}}^{CD_3^+/D-CH_3^+/H}$ also show very small positive delays that all remain below 3 as in magnitude. The experimental results $\tau_{\text{exp}}^{CD_3^+/D-CH_3^+/H}$ are $13 \pm 16$ as and $15 \pm 10$ as at SB12 and SB14, respectively, but also approach 0 as at higher photon energies. As compared to the non-dissociative channels, the one-photon Wigner delay difference $\tau_{\text{Wig}}^{CD_3^+/D-CH_3^+/H}$ strongly departs from both experiments and the full theory below 25 eV. For completeness, we have also determined the angle-resolved

photoionization delays (Fig. S4), but have found no angular dependence within the error bars of our measurements.

We now discuss the interpretation of these new types of photoionization delays, which were made accessible in this work by attosecond coincidence spectroscopy. First, we note the remarkably small isotope effect on the delays $\tau^{CD_4^+ - CH_4^+}$ (Fig. 4a). This is the first important result. As discussed in the introduction, methane displays one of the fastest known structural dynamics following ionization with

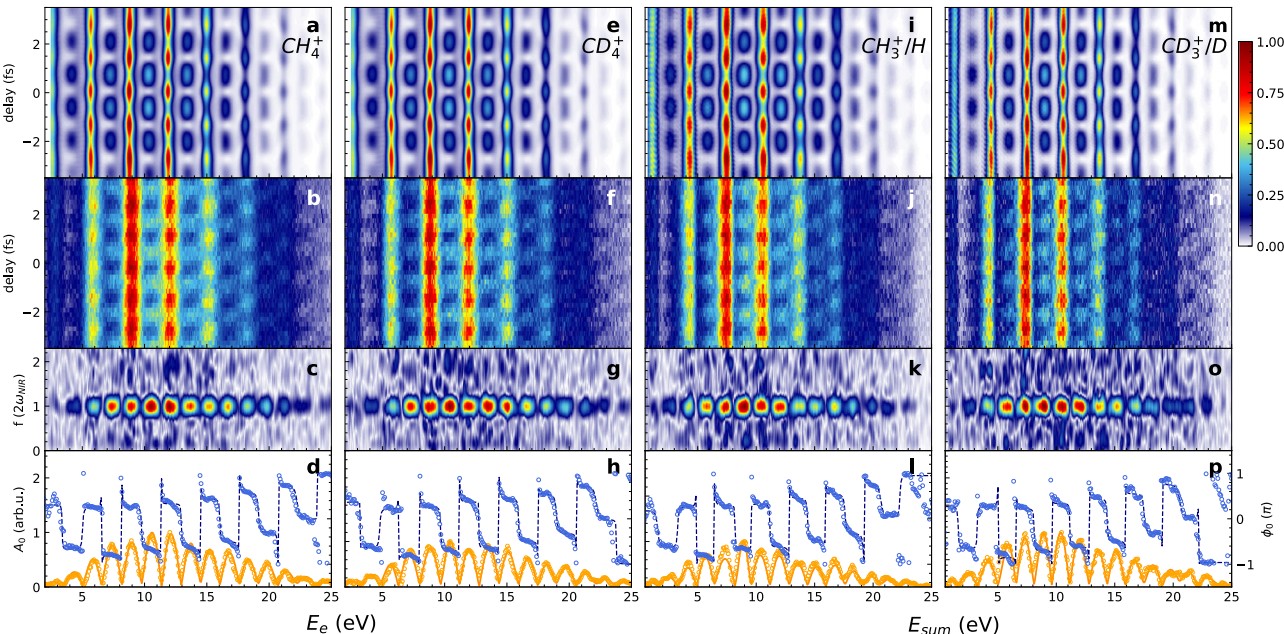

**Fig. 2 | Attosecond photoelectron spectroscopy of methane. a** Theoretically calculated and **b** experimentally measured photoelectron spectrum ($E_e$) of $CH_4$ in coincidence with $CH_4^+$ as a function of APT-IR delay. **c** Fourier-transform amplitude of the attosecond photoelectron spectra in (**b**). **d** Normalized oscillation amplitude ($A_0$) and phase ($\phi_0$) at $2\omega_{NIR}$ of the attosecond photoelectron spectra in (**b**). The experimentally measured $A_0$ is shown as an orange solid line, $\phi_0$ is displayed by the navy dashed line, and the fitted results are shown as colored circles. **e**–**h** Same as (**a**–**d**) but for $CD_4$. **i**–**l** and **m**–**p** show the photoelectron spectra of $CH_4$ and $CD_4$ in coincidence with $CH_3^+/H$ and $CD_3^+/D$ ions, respectively, where $E_{sum}$ indicates the sum of $E_e$ and $E_{mol}$.

vibrational periods as short as 11 fs ($CH_4^+$) and a nuclear auto-correlation function decaying to zero within 2 fs[23]. Yet, an upper limit of less than 10 as is obtained here for $\tau^{CD_4^+ - CH_4^+}$. This indicates that even the fastest nuclear dynamics do not significantly influence molecular photoionization delays in the absence of resonances. It is indeed worth recalling that some effects attributed to nuclear motion have been found in the region of the $Q_1$ resonances of $H_2$[18] and the $3\sigma_g$ shape resonance of $N_2$[10]. In the case of $H_2$, the autoionization lifetimes of the $Q_1$ resonances are on the order of femtoseconds, which provides a long time interval for electron-nuclear coupling to manifest itself. In contrast, in the present study of methane, there is no visible trace of either shape or autoionizing (Feshbach) resonances in the investigated range of photon energies, such that the present work quantifies nuclear-motion effects in the general case of non-resonant photoionization.

In contrast to the (very small) pure isotope effect in $\tau^{CD_4^+ - CH_4^+}$, the delays between dissociative and non-dissociative ionization channels (Fig. 3) are clearly different from zero and the agreement between theory and experiment is outstanding. This provides the opportunity to analyze the origin of such delays in detail. Figure 5a shows a comparison of $\tau^{CH_3^+/H - CH_4^+}$ from the full calculation (green symbols, same data as in Fig. 3a) with the corresponding delays from a calculation that only contains the dominant partial wave ($\ell = 2$, cyan circles). The restriction to the dominant partial wave enables a detailed and transparent analysis of the physical contributions to the delays. Although the green and cyan lines do not quantitatively agree, they are in reasonable agreement for photon energies above 22 eV (corresponding to SB14). We can thus conclude that the simplified calculation containing only the dominant partial wave correctly captures the dominant effects resulting in the observed delays between dissociative and non-dissociative ionization of methane. To understand this effect further, we also performed a full calculation for the one-photon-ionization process in the dissociative and non-dissociative pathways. The grey line shows the calculated Wigner time-delay difference of $\tau_{wig}^{CH_3^+/H - CH_4^+}$, which agrees reasonably well with the full calculation above the photon energy of 25 eV (corresponding to SB16).

We can therefore conclude that the salient features of the time delays between dissociative and non-dissociative ionization are captured by calculations that include the nuclear motion effect, whereby the dominant partial wave is sufficient for a qualitative reproduction of the observed effects. This allows us to further decompose the effect, as shown in Fig. 5b and c. Along the dissociative-ionization pathway, the symmetry of methane is lowered from $T_d$ to $C_{3v}$, which splits the $1t_2$ HOMO into a $1e$ and a $3a_1$ orbital, with the corresponding potential-energy curves shown in Fig. 1b. Whereas the $(1e)^{-1}$ state is strongly bound, the dissociative part of the $(3a_1)^{-1}$ state is also accessible, as illustrated by the Franck-Condon (FC) factors in Fig. 5c. As a consequence, the contribution of the $(1e)^{-1}$ ionization channel is dominated by a single (bound) vibrational level, whereas that of the $(3a_1)^{-1}$ channel includes a range of vibrational levels, as well as the dissociative continuum. Hence, not surprisingly, the difference between one-photon delays associated with the dissociative and non-dissociative channels arising from $(3a_1)^{-1}$ ionization (orange curve in Fig. 5a) is nearly indistinguishable from that given by the grey curve, which includes contributions from both $(1e)^{-1}$ and $(3a_1)^{-1}$ ionization.

Figure 5b then reveals the physical origin of the relative delay between the dissociative and non-dissociative channels. It shows the channel-resolved photoionization delays as a function of the C-H dissociation coordinate, calculated as described in the Methods Section. These delays all monotonically increase from the FC to the dissociated region of the relevant $(3a_1)^{-1}$ channel. Along this dissociative coordinate, the corresponding $3a_1$ orbital increases in its spatial extension, as illustrated in the inset of Fig. 5. Calculations shown in the SM (Fig. S9) demonstrate that the effect of different ionization potentials between the dissociative and non-dissociative ionization channels is irrelevant in the present case. These results, therefore, indicate that the increasing delocalization of the electron wave function along the dissociative-ionization channel is likely to be the main origin of the effects observed in this work. A similar correlation between photo-ionization delays and the spatial extension of electronic wave functions has recently been documented in the case of water clusters[30]. The present results indicate that this effect may also be the cause of the

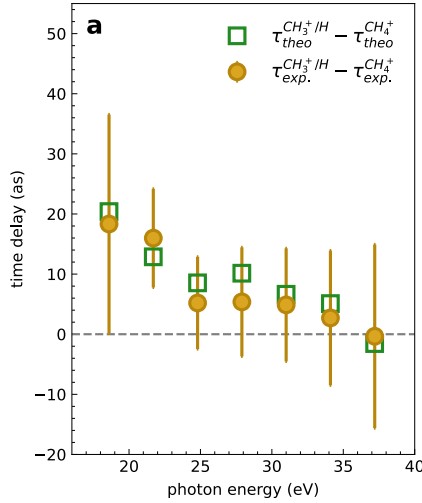

**Fig. 3 | Nuclear motion effect in attosecond photoemission time delays.**
**a** Experimentally measured and theoretically calculated relative photoemission time delays between the dissociative ionization channel $CH_3^+/H$ and the non-dissociative ionization channel $CH_4^+$. **b** Same as (**a**) but for $CD_4$. The error bars represent the standard deviation of the electron sideband within the confidence region of 90%.

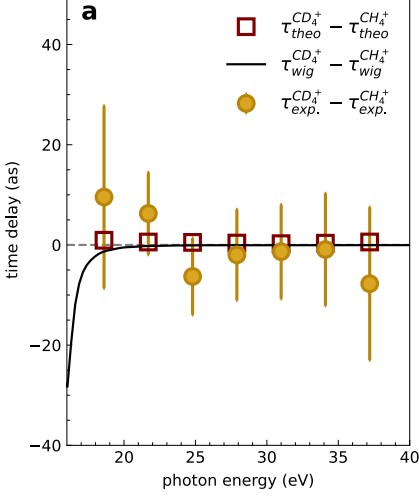

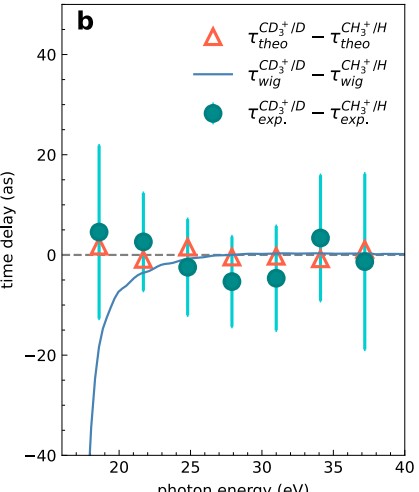

**Fig. 4 | Isotope effect in attosecond photoemission time delays.** Measured and calculated relative photoionization delays between the non-dissociative (**a**) and dissociative (**b**) channels. The one-photon Wigner time delay $\tau_{wig}$ is also shown. The error bars show the standard deviation.

delays observed in the present work, which would offer a conceptually very simple and therefore very powerful predictive framework for photoionization delays in the presence of nuclear motion.

Combining the new opportunities offered by attosecond coincidence spectroscopy with the latest advances in theory, we have performed a detailed characterization of the role of nuclear motion in molecular photoionization delays. We have chosen methane for this study because it features one of the fastest nuclear dynamics following ionization. This study has provided the first evidence of a non-negligible time delay between dissociative and non-dissociative ionization channels. These delays decay from ~20 attoseconds close to the ionization threshold to nearly zero with increasing photon energy and reflect a previously unknown phase shift between the photoelectron continua associated with the dissociative and non-dissociative ionization channels. Remarkably, we have found that the delay differences between $CH_4^+$ and $CD_4^+$ are too small to be measured. Slightly larger delays, albeit still close to zero and within the expected error bars, are measured between the dissociative-ionization channels $CH_3^+/H$ and $CD_3^+/H$. These results offer fundamentally new insights into the role of nuclear motion in attosecond photoionization dynamics. Whereas the

different speeds of nuclear motion caused by deuteration are negligible within a given type of ionization channel, nuclear motion can have a measurable effect on the relative delays between channels involving different types of nuclear motion. This offers interesting new perspectives for the study of coupled electron-nuclear dynamics on attosecond time scales[17,33], such as the comparison of the effects of adiabatic versus non-adiabatic crossings of conical intersections[34] on photoionization delays.

## Methods

### Attosecond coincidence spectroscopy

The experiments have been performed using an attosecond coincidence interferometer including a phase-locked XUV-APT and NIR pump-probe beamline and a COLTRIMS. A regeneratively amplified Ti: Sapphire laser system is used to generate 1.2-mJ NIR pulses with a central wavelength of 800 nm and a full-width-at-half-maximum pulse duration of 35 fs at a 5 kHz repetition rate. The NIR pulses are separated into two arms through a beam splitter and 70% of the laser beam is focused into the high-harmonic-generation (HHG) cell, which is filled with xenon or argon for HHG with the XUV photon energies up to H13

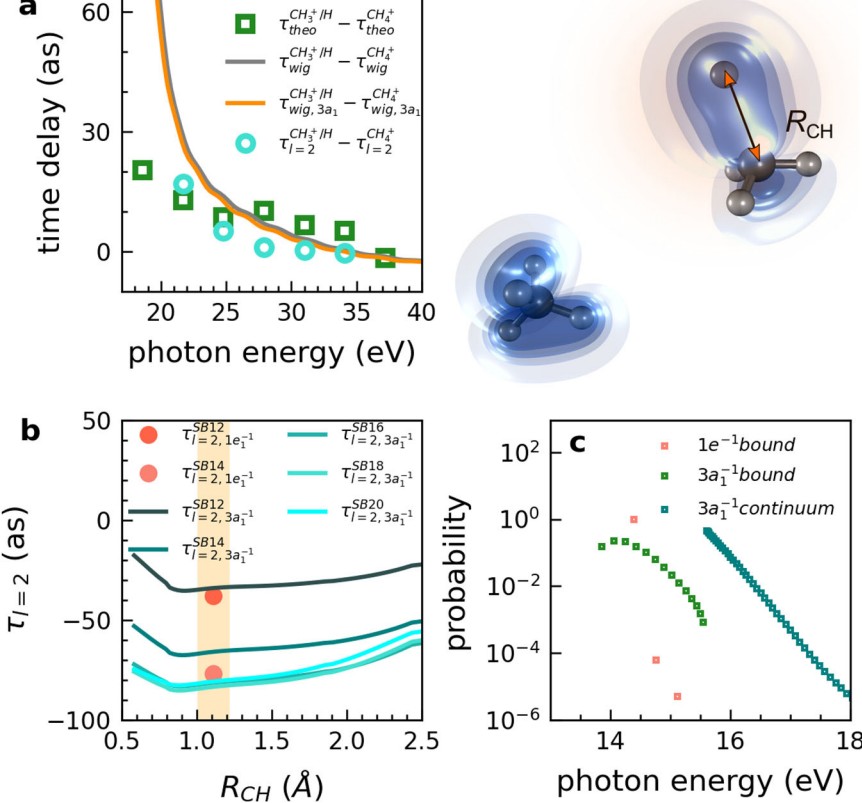

**Fig. 5 | Nuclear motion effect of spatial delocalization in attosecond photo-emission time delays. a** Calculated relative photoemission delays between the dissociative ($CH_3^+/H$) and non-dissociative channel ($CH_4^+$) as a function of the photon energy, obtained from the complete calculation, $\tau_{theo}^{CH_3^+/H} - \tau_{theo}^{CH_4^+}$ (green squares, same as in Fig. 3a), from only considering the dominant partial wave $l = 2$, $\tau_{l=2}^{CH_3^+/H} - \tau_{l=2}^{CH_4^+}$ (cyan circles), from the difference between the corresponding one-photon Wigner time delays, $\tau_{wig}^{CH_3^+/H} - \tau_{wig}^{CH_4^+}$ (grey line), and from the difference between the corresponding one-photon Wigner time delays but only considering the 3a₁ channel, $\tau_{wig,3a_1}^{CH_3^+/H} - \tau_{wig,3a_1}^{CH_4^+}$ (orange curve). The latter two delay differences have been evaluated as an average over vibrational and dissociative states (see SM). The right panel shows the electron density distribution of the $3a_1$ orbital along the C-H bond stretching from 1.1 Å to 2.5 Å in $C_{3v}$ symmetry. **b** Photoionization delays of SB12-20, $l = 2$, for ionization from the 3a₁ or 1e orbitals, as a function of the C-H internuclear separation. The orange-shaded area indicates the Frank-Condon region. **c** Symmetry-resolved Franck-Condon factors as a function of the photon energy.

or H25, respectively. A 100-nm thick aluminum foil is inserted in front of the nickel-coated toroidal mirror ($f = 50$ cm) to filter out the fundamental NIR beam and further compress the chirp of the XUV-APT. A much weaker NIR pulse from the residual part of the laser beam is recombined with the XUV-APT through a hole mirror to build up a nonlinear Mach-Zehnder interferometer, and the relative path length is actively stabilized through a piezoelectric motor with a time jitter of less than 40 as. The phase-locked XUV-APT and NIR pulses are focused into the supersonic gas jet of a mixture of $CH_4$ (40%), $CD_4$ (40%), and Ar (20%) in an ultrahigh vacuum chamber (below $4.99 \times 10^{-10}$ mbar) of the COLTRIMS. The photoelectrons and -ions are guided by homogeneous electric and magnetic fields onto the position- and time-of-flight-sensitive detectors on opposite sides of the spectrometer. The field parameters are 2.5 V/cm and 5.8 Gauss in the argon HHG conditions, and 2 V/cm and 5.2 Gauss in the case of xenon HHG. Both the XUV-APT and NIR pulses are vertically polarized along the z-axis, i.e., the direction of the spectrometer and the perpendicular laser and gas-jet propagation directions are along x- and y-axes, respectively. The intensity of the NIR pulse at the interaction region is estimated to be ~$10^{12}$ W/cm². Details on the data analysis are given in the SM (Section 2).

## Theory
Vibrationally and fragment kinetic-energy resolved RABBIT spectra have been obtained by solving the $N$-electron time-dependent Schrödinger equation (TDSE) in which nuclear motion leading to

the breakup of one of the C-H bonds is accounted for in an adiabatic manner (see SM, Section 3 for details). The nuclear components of the wave packet are obtained by solving the nuclear time-independent Schrödinger equation in a cut of the potential-energy surface along the C-H coordinate leading to dissociation, obtained at the CASSCF/MRCI level. The multiconfigurational nature of these calculations ensures an accurate description of the dissociative asymptotes, as demonstrated in Table S1. The $N$-electron component of the wave packet, properly accounting for the electronic continua, are obtained by using an extension of the static-exchange density functional theory (static-exchange DFT) method described in[35] (see SM, Section 3). The TDSE is then solved for the combined action of an attosecond pulse train (APT) and the 800-nm NIR field used to generate the former through the HHG process, thus accounting for all bound-bound, bound-continuum, and continuum-continuum transitions produced in the RABBIT experiments. The chosen APT and NIR fields have peak intensities of $10^{11}$ and $10^{12}$ W/cm², respectively, with perfectly Gaussian envelopes and a full-width half-maximum (FWHM) of 8.6 fs, which corresponds to a total NIR pulse duration of around 22 fs. Additional details are given in the SM (Section 3).

## Data availability
The data that support the findings of this study are saved in the online repository https://doi.org/10.3929/ethz-b-000600846.

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

## Acknowledgements

We thank A. Schneider and M. Seiler for technical support. We thank Prof. Piero Decleva for insightful discussions in the early stages of this project and for technical advice on the use of the static exchange DFT methodology. We gratefully acknowledge funding from an ERC Consolidator Grant (Project Nos. 772797-ATTOLIQ), the SNSF through grant number 200021_172946, the European COST Action CA18222 Atto-Chem, the Ministerio Español de Ciencia e Innovación (MICINN) through the projects PID2019-105458RB-I00, the Severo Ochoa Programme for Centres of Excellence in R&D (CEX2020-001039-S) and the María de Maeztu Programme for Units of Excellence in R&D (CEX2018-000805-M), the Comunidad de Madrid through the projects FULMATEN Ref. Y2018NMT-5028 and PRICIT Ref. PCD-I3PCD026, as well as the NCCR-MUST, funding instrument of the Swiss National Science Foundation, the National Natural Science Foundation of China (Grants Nos. 12122404,11974114,12261160363), the Shanghai Science and Technology Commission (Grant No.19560745900), and the Fundamental Research Funds for the Central Universities. Computational resources provided by the MareNostrum Supercomputer Centre through the Spanish Supercomputing Network and the Scientific Computation Center at the Universidad Autónoma de Madrid (CCC-UAM) are acknowledged.

## Author contributions

H.J.W. and X.G. conceived the idea and initiated the study. X.G. and S.H. carried out the experiments and the data analysis. X.G. and H.J.W. interpreted the experimental results. E.P., A.P., and F.M. performed the

theoretical calculations. X.G. and H.J.W. wrote the paper with input from all co-authors.

## Competing interests

The authors declare no competing interests.
