## [Peer Review File · Nature Communications]

Attosecond delays between dissociative and non-dissociative ionization of polyatomic moleculesREVIEWER COMMENTS

Reviewer #1 (Remarks to the Author):

This is clearly a complete study of time delays in molecular ionization-dissociation where isotopic effects should appear.

The combination of expert theory (Martin) and experiments (Worner) is commendable and supports the publishable recommendation.

Nevertheless a few comments would help the paper to be more appreciated.

In the abstract one is told that the paper deals with the fastest nuclear motion however in the text electron and nuclear time scales are not clearly identified. In particular ultrafast nondissociative excitation and ionization will occur on attoseconds. Furthermore such excitations will create nonadiabatic transitions involving electron localization at different nuclear distances, which should create strong differences in CH⁺ and CD⁺ signals. It was surprising to find that there are often no differences in these isotope effects which are strongly sensitive to attosecond electron localization, an effect not addressed seriously but now considered important as in PRA 104,043101(2021) by F He.

Finally the use of static exchange TDDFT is a serious limitation in treating time dependent attosecond electron localization. Recent work by N Maitra, PRA 104,03281(2021) points out the deficiencies of TDDFT, which neglect virtual time dependent effects.

The paper clearly requires some explanations and justifications of the neglect of the above mentioned physical effects.

Reviewer #2 (Remarks to the Author):

The authors reported on the experimental and theoretical study of photoionization dynamics in polyatomic molecules (CH₄ and CD₄). By combining the RABBIT measurement and coincidence technique, the photoionization time delay between dissociative and non-dissociative ionization channels in a single molecule, or between different isotopic molecules can be measured with attosecond resolution. The measurements indicate that no isotope relative time delay is found between CH₄ and CD₄. However, a 20 as time delay between the dissociative and non-dissociative ionization of both CH₄ and CD₄ is observed. The N-electron TDSE model including the nuclear motion gives reasonable agreement with the experimental data. The main findings are 1) the fast nuclear dynamics do not significantly alter the photoionization delay; 2) the increasing delocalization of the electron wave function during dissociation is likely responsible for the origin of the measurable delay between the dissociative and non-dissociative channel. It is an elegant experimental work and is beneficial for quantitatively studying the XUV photoionization dynamics in molecules. However, the effect of the

nuclear motion on photoionization phase (or delay) has been reported in Ref[18], in which the identical technique has been utilized to explore similar physical processes. Given the work of ref [18] the significance and novelty of the current work is remarkably reduced. The current work is more of a marginal extension of Ref [18] in a less systematic way for different molecules. The photoelectron spectrum is averaged over all emission angles, this could cause apparent discrepancy between theory and experiment. I am surprised that the authors can still claim to get outstanding agreement between theory and experiment. As comparison ref [18] measured the photoelectron spectra in an angle-resolved way and provides a better analysis framework. My impression is that it is difficult to gain fundamentally new information either technically or scientifically from the current work. Therefore, I do not think it meets the publication criteria of nature communications.

Other comments:

1. I would expect a narrower energy conservation line between electron and ion in Figure 1E. A broader distribution can be either due to the inclusion of multiple cation states or false coincidence events, which have to be clarified very carefully before any reasonable interpretation of the data.

2. This is associated with previous comment. I think a reliable energy sharing map requires hours' run to get sufficient statistics, this could be very challenging for the RABBITT measurement of the dissociative channels. The authors should provide more detailed information of the data acquisition procedure, and also the energy sharing maps at different delays should be provided to support their conclusion.

3. How are the error bars in Fig3 and Fig4 defined?

Reviewer #3 (Remarks to the Author):

Let me begin by saying that this manuscript is very well written. That assessment concerns the words, not the figures, as I will explain below.

I find the topic interesting, and I do support acceptance for Nature Communications, in spite of the fact that the error bars on ALL experimental points presented in Figs. 3 and 4 are consistent with zero! In other words, the actual conclusions about the relevant physics/chemistry are really based on the theory. I am not an expert regarding the details of the numerical approach, but it does appear to me that this is a state-of-the-art calculation. In fact, for full disclosure, I do know the senior theorists on the author list, and I have a very high opinion of them.

I also believe that, in spite of the long error bars (I appreciate that the authors reported them in their full length, even though reviewers like myself may pick up on it) that this is a state-of-the-art experiment. Given the error bars, one cannot really say that the experimental data validate the predictions, only that the data do not contradict them.

I also have many comments about the figures. Some of them may be a matter of taste, although I do believe that when I learned these rules (from experimentalists, for that matter) that they make sense. I'll start with the main manuscript:

1) I do not see any reason to connect the data points (both experiment and theory) by a line in Figs. 3 and 4, and certainly not by straight lines. The one-and-only justification may be to "guide the eye", but that is not necessary either. The symbols are clear enough.

BUT: Given the importance of "zero" (or "non-zero") in this work, I strongly recommend to add a zero line in the panels of Figs. 3 and 4.

2) The very yellow color of the experimental error bars in Figs. 3A and 4A does not match the symbol color well. In fact, yellow is generally not recommended anyway. Why not use black?

3) I would prefer to just have the symbols in the legend (even for the experimental data), rather than an indication of a horizontal error bar when all the data are presented with vertical error bars. I realize that this is likely a "feature" (unfortunately, a bad one) of the authors' graphics program. It would require some extra work to improve on this, but if one just uses the symbols, one could simply add in the figure caption that the experimental points are shown with vertical error bars.

4) There is really no need to have a circle with what looks like a horizontal error bar in the legend of Fig. 5 B, when there are only two circles for the actual results.

5) The "curves" in the panels of Fig. 5 look too rugged to me. Again, I don't think that there is a need to connect the squares in panel A by lines. For all the other curves, I would try a spline interpolation rather than what looks again like a straight line.

6) And, finally, my eyes would like it (a lot) better if "A", "B", and "C" were put INSIDE the frames rather than at the top left of the entire panel. In Figs. 3 and 4, this could easily be done in the top left of the frame, while some different choices may be needed in the panels of Fig. 5. In fact, do they really have to be capitalized and in bold face?

Now on to the Supplementary Materials:

Fig. S2: Many of the letters are very (too?) small.

Fig. S4: I would move the legend box outside of the frame (to the right). There is so much space. No need to cover up 20% of the curves!

Fig. S5: Same as for S4.

Fig. S6: See previous comments on the style and positioning of "A" and "B". [I do, however, applaud the fact that the experimental data are NOT connected by a line.]

Fig. S7: See previous comments on "A" and "B".

Fig. S8: See previous comments on "A" and "B". And, once again, I do not think that connecting the points by straight lines is appropriate. Either drop the lines or use a better interpolation scheme.

Fig. S9: Are the wiggles and sharp structures in two of the curves (one just below, the other just above 20 eV) real physics/chemistry? Did I miss something?

Fig. S10: Fig. S8: See previous comments on "A", "B", ...

To summarize: I support publication in Nature Communications, because I believe that this work represents an excellent effort and the results are very interesting. I won't insist on improving the figures, but I do believe that excellent work deserves excellent presentation. For the present work, I judge the words as very good, but there is definitely room for improvement in the figures.

Klaus Bartschat

We thank all three reviewers for their detailed reading of our manuscript and their thoughtful comments. We have revised the manuscript according to their recommendations. Below, we reproduce their report in black/red, provide our answers in blue and the changes made to the manuscript in green.

Reviewer #1 (Remarks to the Author):

This is clearly a complete study of time delays in molecular ionization-dissociation where isotopic effects should appear.

The combination of expert theory (Martin) and experiments (Worner) is commendable and supports the publishable recommendation.

We thank the reviewer for their supportive comments.

Nevertheless a few comments would help the paper to be more appreciated.

In the abstract one is told that the paper deals with the fastest nuclear motion however in the text electron and nuclear time scales are not clearly identified. In particular ultrafast nondissociative excitation and ionization will occur on attoseconds. Furthermore such excitations will create nonadiabatic transitions involving electron localization at different nuclear distances, which should create strong differences in CH⁺ and CD⁺ signals. It was surprising to find that there are often no differences in these isotope effects which are strongly sensitive to attosecond electron localization, an effect not addressed seriously but now considered important as in PRA 104,043101(2021) by F He.

We appreciate the reviewer's recommendations for improving the manuscript. We were also surprised to find that the isotope effects on the photoionization delays were so small. We are also aware of the significance of these isotope effects in molecular hydrogen. However, the strong non-adiabatic transitions described in the mentioned work are the consequence of the small energy gaps between the two lowest electronic states in molecular hydrogen. In the case of methane cation, this situation does not arise because the lowest-lying electronically excited state ($2a_1^{-1}$) of the methane cation lies more than 8 eV above its electronic ground state ($1t_2^{-1}$) [W. Potts and W. C. Pryce, Proc. R. Soc. Lond. 326, 165 (1976)]. Moreover, these two electronic states do not share a common dissociation threshold. For these reasons, we do not expect significant non-adiabatic transitions to occur between these two electronic states and we therefore do not expect electron localization to take place during the ionization of methane.

We have added the following information to the main text (p. 7):

"As discussed in the introduction, methane displays one of the fastest known structural dynamics following ionization with vibrational periods as short as 11~fs (CH₄⁺) and a nuclear auto-correlation function decaying to zero within 2 fs [cite{mondal14a}]."

Finally the use of static exchange TDDFT is a serious limitation in treating time dependent attosecond electron localization. Recent work by N Maitra, PRA 104,03281(2021) points out the deficiencies of TDDFT, which neglect virtual time dependent effects.

The paper clearly requires some explanations and justifications of the neglect of the above mentioned physical effects .

We agree with the referee that TDDFT, which is a single Slater determinant approach in which the electronic density is written in terms of Kohn-Sham DTF orbitals, has problems to describe electron localization. These problems mainly arise when two nearby dissociative states are very close in energy and are efficiently mixed by non-adiabatic transitions or an external field, in which case one has to consider at least two Slater determinants (one per electronic state). We would like to clarify however that, strictly speaking, we are not using TDDFT. We use static-exchange DFT to describe the molecular orbitals of CH₄, both bound and continuum molecular orbitals, in a purely stationary picture. Then we use these orbitals as a basis to solve the N-electron time-dependent Schrödinger equation (TDSE), where N is the number of electrons in CH₄, assuming that the corresponding N-electron time-dependent wave function can be written as a single Slater determinant. Therefore, we are not propagating in time the electron density, as in TDDFT, but a single-Slater determinant **wave function** built from one-electron orbitals that depend on time. In this sense, our method is closer to time-dependent Hartree-Fock (TDHF) than to TDDFT. Nevertheless, as our method is also a single Slater determinant approach, it may also be affected by the limitations pointed out by the referee, e.g., its inability to describe electron localization when two different states are close in energy or tend to the same dissociation limit. However, as mentioned in our answer to the previous point, such localization effects are irrelevant in the present case due to the large energy separation between the states leading to dissociation, even at infinite separation between the dissociation products.

As this is an interesting point that is worth clarifying to support the validity of our calculations, we have included the following sentence in the supplementary material (p. 8):

“We note that, although this is a single Slater determinant approach, which may be inappropriate to describe electron localization, which usually implies the mixing of several Slater determinants [N Maitra, PRA 104, 03281(2021)], this is not a real limitation in the present case, since the energy separation between the states involved in the dissociative ionization processes is very large, even at infinite separation between the dissociation products.”

Reviewer #2 (Remarks to the Author):

The authors reported on the experimental and theoretical study of photoionization dynamics in polyatomic molecules (CH₄ and CD₄). By combining the RABBIT measurement and coincidence technique, the photoionization time delay between dissociative and non-dissociative ionization channels in a single molecule, or between different isotopic molecules can be measured with attosecond resolution. The measurements indicate that no isotope relative time delay is found between CH₄ and CD₄. However, a 20 as time delay between the dissociative and non-dissociative ionization of both CH₄ and CD₄ is observed. The N-electron TDSE model including the nuclear motion gives reasonable agreement with the experimental data. The main findings are 1) the fast nuclear dynamics do not significantly alter the photoionization delay; 2) the increasing delocalization of the electron wave function during dissociation is likely responsible for the origin of the measurable delay between the dissociative and non-dissociative channel.

It is an elegant experimental work and is beneficial for quantitatively studying the XUV photoionization dynamics in molecules.

We thank the reviewer for the accurate summary and for acknowledging the elegance of our work.

However, the effect of the nuclear motion on photoionization phase (or delay) has been reported in Ref[18], in which the identical technique has been utilized to explore similar physical processes. Given the work of ref [18] the significance and novelty of the current work is remarkably reduced. The current work is more of a marginal extension of Ref [18] in a less systematic way for different molecules.

We thank the referee for making us realize that we have not sufficiently clarified the differences between our work and that of Ref. [18]. Beyond the common experimental technique, there is not much similarity between our work and Ref. [18]. The work of ref. [18] is on H₂ (only) and the effects of nuclear motion were exclusively explored in the vicinity of Feshbach resonances, where they are expected to be unusually large due to the long autoionization time (several femtoseconds). These long lifetimes allow the nuclei to move substantially during the autoionization process, which consequently leads – unsurprisingly – to large nuclear-motion effects.

In the present case, we do not have any (Feshbach) resonances. Instead, we are studying direct ionization delays, which are on the order of tens of attoseconds, i.e., at least two orders of magnitude faster. And in spite of this ~100-times faster ionization process, we do observe an effect due to nuclear motion, which is small, but we quantify it. This quantitative measurement of nuclear-motion effects in regular (non-resonant) photoionization is unprecedented and pushes the limits in the required precision of experiments and calculations to a new level. Furthermore, to our knowledge, this is the first time that differences in the delays between dissociative and non-dissociative ionization processes have even been measured and calculated in a polyatomic molecule.

To clarify the differences relative to Ref. [18], we have added/rewritten the following text: (p.8) “In the case of H₂ the autoionization lifetimes of the Q1 resonances are on the order of femtoseconds, which provides a long-time interval for electron-nuclear coupling to manifest itself. In contrast, in the present study of methane, there is no visible trace of either shape or autoionizing Feshbach resonances in the investigated range of photon energies, such that the present work quantifies nuclear-motion effects in the general case of non-resonant photoionization.”

The photoelectron spectrum is averaged over all emission angles, this could cause apparent discrepancy between theory and experiment. I am surprised that the authors can still claim to get outstanding agreement between theory and experiment.

We have to admit that we cannot follow the logic of this comment. First, let us point out that both experiment and theory are averaged over emission angles and that the averaging is done in the same way. Second, the outstanding agreement between theory and experiment is not only a claim, it is a fact. Since we are comparing measured angle-averaged delays to calculated angle-averaged delays, obtained according to the same averaging procedure as the experimental data, the agreement reflects the quality of the experimental and theoretical work. Finally, averaging over emission angles cannot be expected to make agreement between experiment and theory worse. Indeed, in general, the more differential a measurement the more challenging to reproduce by theory, since errors due to small deficiencies in the calculated wave functions may compensate when performing the averaging while they can be quite visible in differential observables. For example, there are numerous works showing that calculations of molecular-frame photoelectron angular distributions are not so well converged in cases where angle integrated cross sections are totally converged.

As comparison ref [18] measured the photoelectron spectra in an angle-resolved way and provides a better analysis framework. My impression is that it is difficult to gain fundamentally new information either technically or scientifically from the current work. Therefore, I do not think it meets the publication criteria of nature communications.

This study reports information that is not only technically and scientifically new, but even unprecedented. This work is the first to compare dissociative and non-dissociative ionization channels, which was not done in Ref. [18]. This allows us to unequivocally observe signatures of nuclear motion in an ionization process that only takes a few attoseconds to occur! This is much more challenging than studying the effect of nuclear motion when ionization occurs in the vicinity of (Feshbach) resonances (as in Ref. 18), which takes several femtoseconds (not attoseconds). The case considered in this work is the rule rather than the exception in molecular photoionization at high photon energies. Our work is also the first to compare photoionization delays of two isotopic species in the presence of *fast* nuclear dynamics, which has not been done in Ref. [18]. Finally, let us point out that the angle dependence of the photoionization delays is not the topic of this work. We do indeed not expect any significant angle-dependence on the nuclear-motion effects studied in this work. Therefore, there was no need and no point in measuring (or calculating) the photoionization delays as a function of emission angle.

Other comments:

1.I would expect a narrower energy conservation line between electron and ion in Figure 1E. A broader distribution can be either due to the inclusion of multiple cation states or false coincidence events, which have to be clarified very carefully before any reasonable interpretation of the data.

We thank the referee for sharing these concerns with us. In contrast to diatomic molecules, methane has a very broad photoelectron spectrum (see Fig. 1D and the literature, i.e. W. Potts and W. C. Pryce, Proc. R. Soc. Lond. 326, 165 (1976)). Unlike the referee, we are therefore not surprised by the width of the energy conservation line. Additionally, we have of course carefully excluded the contributions of false coincidences to the presented results. Specifically, we have reduced the target density and increased the distance between the nozzle and laser focus position in the center of the reaction microscope to maintain the ion and electron events at 0.15

and 0.30 count per laser shot. We have ensured that this eliminates the contributions of false-coincidences by recording photoelectron spectra from mixtures of Ar and CH₄, where the photoelectron signals from the two species can be clearly distinguished. This confirmed the absence of false coincidences.

2. This is associated with previous comment. I think a reliable energy sharing map requires hours' run to get sufficient statistics, this could be very challenging for the RABBITT measurement of the dissociative channels. The authors should provide more detailed information of the data acquisition procedure, and also the energy sharing maps at different delays should be provided to support their conclusion.

We agree with the referee that the attosecond coincidence measurement of the dissociation channel is challenging. However, we have succeeded in recording such maps at many XUV-IR delays, thanks to the exceptional long-term stability of our measurements. As shown in Fig. 2j and 2n, the spectrum of E_{sum} presents a clear sideband and main-peak modulation as a function of the XUV-IR delay.

We have added a new Fig. S2, which presents the time-resolved joint energy spectrum between electron and nuclei in the dissociation channel CH₃⁺/H, in which the relative intensity of the sidebands and main peaks can clearly be seen to oscillate as a function of the pump-probe time delay.

3. How are the error bars in Fig3 and Fig4 defined?

The error bars in Fig. 3 and Fig. 4 are both calculated of the standard deviation of the electron sideband within the confidence region of 90%.

To make it clear to readers, we have clarified the error bar presentation of Fig. 3 and Fig. 4 in the resubmitted manuscript.

Reviewer #3 (Remarks to the Author):

Let me begin by saying that this manuscript is very well written. That assessment concerns the words, not the figures, as I will explain below.

We thank the reviewer for their supportive comments.

I find the topic interesting, and I do support acceptance for Nature Communications, in spite of the fact that the error bars on ALL experimental points presented in Figs. 3 and 4 are consistent with zero! In other words, the actual conclusions about the relevant physics/chemistry are really based on the theory.

I am not an expert regarding the details of the numerical approach, but it does appear to me that this is a state-of-the-art calculation. In fact, for full disclosure, I do know the senior theorists on the author list, and I have a very high opinion of them.

I also believe that, in spite of the long error bars (I appreciate that the authors reported them in their full length, even though reviewers like myself may pick up on it) that this is a state-of-the-art experiment. Given the error bars, one cannot really say that the experimental data validate the predictions, only that the data do not contradict them.

We are grateful to the reviewer for their kind words. However, we would like to point out that the measured delays between dissociative and non-dissociative photoionization channels (Fig. 3) are not (all) consistent with zero. Most importantly, the measured and calculated delays display the same trend as a function of kinetic energy and the agreement between theory and experiment is quantitative, i.e. the calculated delays lie well within the experimental error bars.

I also have many comments about the figures. Some of them may be a matter of taste, although I do believe that when I learned these rules (from experimentalists, for that matter) that they make sense. I'll start with the main manuscript:

1) I do not see any reason to connect the data points (both experiment and theory) by a line in Figs. 3 and 4, and certainly not by straight lines. The one-and-only justification may be to "guide the eye", but that is not necessary either. The symbols are clear enough.

BUT: Given the importance of "zero" (or "non-zero") in this work, I strongly recommend to add a zero line in the panels of Figs. 3 and 4.

We agree with the reviewer.

We have removed the connecting lines and add the "zero" lines in Figs. 3 and 4.

2) The very yellow color of the experimental error bars in Figs. 3A and 4A does not match the symbol color well. In fact, yellow is generally not recommended anyway. Why not use black?

We thank the reviewer for his kind suggestions.

The Figs.3 and 4 have been modified accordingly.

3) I would prefer to just have the symbols in the legend (even for the experimental data), rather than an indication of a horizontal error bar when all the data are presented with vertical error bars. I realize that this is likely a "feature" (unfortunately, a bad one) of the authors' graphics program. It would require some extra work to improve on this, but if one just uses the symbols, one could simply add in the figure caption that the experimental points are shown with vertical

error bars.

We thank the reviewer for his kind suggestions. The horizontal line issue is fixed in the new legends.

4) There is really no need to have a circle with what looks like a horizontal error bar in the legend of Fig. 5 B, when there are only two circles for the actual results.

Indeed, we have updated Fig. 5B.

5) The "curves" in the panels of Fig. 5 look too rugged to me. Again, I don't think that there is a need to connect the squares in panel A by lines. For all the other curves, I would try a spline interpolation rather than what looks again like a straight line.

We thank the reviewer for these kind suggestions.

Figure 5 has been fully updated in the resubmitted manuscript.

6) And, finally, my eyes would like it (a lot) better if "A", "B", and "C" were put INSIDE the frames rather than at the top left of the entire panel. In Figs. 3 and 4, this could easily be done in the top left of the frame, while some different choices may be needed in the panels of Fig. 5. In fact, do they really have to be capitalized and in bold face?

We have checked the formatting requirements of Nat. Comm. and have accordingly moved the labels inside the frames of the figures and have changed them to lower-case letters.

Now on to the Supplementary Materials:

Fig. S2: Many of the letters are very (too?) small.

Indeed, we have increased the font size.

Fig. S4: I would move the legend box outside of the frame (to the right). There is so much space. No need to cover up 20% of the curves!

Fig. S5: Same as for S4.

These figures have been updated.

Fig. S6: See previous comments on the style and positioning of "A" and "B". [I do, however, applaud the fact that the experimental data are NOT connected by a line.]

We thank the reviewer for his kind suggestions.

Fig. S6 has been updated.

Fig. S7: See previous comments on "A" and "B".

Fig. S7 has been updated.

Fig. S8: See previous comments on "A" and "B". And, once again, I do not think that connecting the points by straight lines is appropriate. Either drop the lines or use a better interpolation scheme.

Fig. S8 has been updated.

Fig. S9: Are the wiggles and sharp structures in two of the curves (one just below, the other just above 20 eV) real physics/chemistry? Did I miss something?

Fig. S9 has been updated.

Fig. S10: Fig. S8: See previous comments on "A", "B", ...

Fig. S10 has been updated.

To summarize: I support publication in Nature Communications, because I believe that this work represents an excellent effort and the results are very interesting. I won't insist on improving the figures, but I do believe that excellent work deserves excellent presentation. For the present work, I judge the words as very good, but there is definitely room for improvement in the figures.

Klaus Bartschat

We thank Prof. Bartschat for his excellent recommendations.

REVIEWER COMMENTS

Reviewer #1 (Remarks to the Author):

I would recommend the authors help readers to believe the interpretation. Thus at the end of the response to reviewer # 1, energy separations and identification of the important states should be added to convince readers since at large distances single determinants are a serious limitation in theoretical interpretation of dissociation. The authors should point out how to avoid this complication in order to make their interpretation more acceptable.

Reviewer #2 (Remarks to the Author):

I have reviewed the revised manuscript and the response to my previous report. While the authors have addressed some of my concerns, I still believe that the novelty and significance of the current work are not convincingly addressed. My main concerns are:

1. Observing the nuclear effect in a faster process as compared to a slower process using the common experimental method does not indicate clear advancement technically, only shows the versatility of the method. The statement "The case considered in this work is the rule rather than the exception in molecular photoionization at high photon energies" is claiming that the "non-resonant ionization" is more general and important to govern the rule over "near resonance ionization", which is absolute not an acceptable statement. Therefore, I still do not see clear significance and novelty of the current work given the work of ref[18].

2. In their reply, the authors stated that "Finally, let us point out that the angle dependence of the photoionization delays is not the topic of this work". As shown in ref 18, measuring the photoelectron spectra in an angle-resolved way can provide a better analysis framework and a more comprehensive view of the dynamics. Therefore, "the angle dependence of the photoionization delays" may not be the topic of this work, but "the angle-resolved measurement of the photoelectron spectra" should be a significant part of the measurement to support the conclusion.

3. In their reply, the authors stated that "We do indeed not expect any significant angle-dependence on the nuclear-motion effects studied in this work. Therefore, there was no need and no point in measuring (or calculating) the photoionization delays as a function of emission angle." I find this statement somewhat disappointing. If the authors wanted to show the "rule" as they indicated in the reply, they should perform a convincing study to confirm the "no need and no point" from their expectation (which is not so obvious to me) rather than throwing away doubts and suggestions.

In sum, while the authors have addressed some of my concerns, I still do not see clear significance and novelty in the current work. Therefore, my original decision remains unaltered.

We thank both referees for agreeing to review our manuscript a second time and for their thoughtful comments. We have revised the manuscript according to their recommendations. We trust that our manuscript is now publishable.

Below, we reproduce the referee comments in black, provide our answers in blue and detail the changes made to the manuscript in green.

Reviewer #1 (Remarks to the Author):

I would recommend the authors help readers to believe the interpretation. Thus at the end of the response to reviewer # 1, energy separations and identification of the important states should be added to convince readers since at large distances single determinants are a serious limitation in theoretical interpretation of dissociation. The authors should point out how to avoid this complication in order to make their interpretation more acceptable.

We thank the reviewer for this comment. It is however important to point out that there is no problem with the energetics, since although we are using a single-determinant approach for the time propagation of the electronic wave function, we are using a full CI methodology to calculate the potential energy curves that we use to obtain the vibrational wave functions. This is clearly explained in the SM, Section 3. Whereas it is indeed necessary to use the full CI methodology to obtain the correct asymptotic behavior of the potential energy curves, and therefore the correct vibrational wave functions, this methodology can neither be used for the time propagation of the electronic wave function in the present work, nor is it necessary, because of the very limited excursion of the vibrational wave packet from the Franck-Condon region during the process of photoionization, which is the subject of our work.

To clarify these aspects further, we have added:

- 1. A table of the dissociation limits in the limit to validate the quantitative accuracy of our results in the Supplementary Material (Table S1).
- 2. The following text in the main manuscript (p. 11): "The multiconfigurational nature of these calculations ensures an accurate description of the dissociative asymptotes, as demonstrated in Table S1."

Reviewer #2 (Remarks to the Author):

I have reviewed the revised manuscript and the response to my previous report. While the authors have addressed some of my concerns, I still believe that the novelty and significance of the current work are not convincingly addressed. My main concerns are:

1. Observing the nuclear effect in a faster process as compared to a slower process using the common experimental method does not indicate clear advancement technically, only shows the versatility of the method. The statement "The case considered in this work is the rule rather than the exception in molecular photoionization at high photon energies" is claiming that the "non-resonant ionization" is more general and important to govern the rule over "near resonance ionization", which is absolute not an acceptable statement. Therefore, I still do not see clear significance and novelty of the current work given the work of ref[18].

Reference [18] has shown, through comparison with advanced theory, that nuclear-motion effects are important in the special case of near-resonant photoionization of H₂ in the vicinity of the Q₁ resonances with lifetimes in the few-femtosecond regime. The present work shows that nuclear-motion effects are negligible in the case of non-resonant (attosecond), non-dissociative ionization, and this conclusion is reached already on the basis of the purely experimental comparison of CH₄ and CD₄. It is simply a fact that non-resonant ionization is more general than near-resonant ionization because

resonances are molecule-specific and only found in the vicinity of ionization thresholds, whereas non-resonant ionization occurs over very broad spectral ranges.

The present work additionally shows that nuclear-motion effects in non-resonant (attosecond), dissociative ionization can be significant (up to 20 as) and this result again follows from the experimental comparison of the dissociative channels of CH₄ and CD₄ directly. Comparison to advanced theory *quantitatively* confirms both this observation and the absence of nuclear-motion effects in the non-dissociative ionization case. This level of agreement between experimental and theoretical delays in molecular photoionization is unprecedented. Theory moreover allows us to rationalize both our experimental observations in terms of the phase of the ionization matrix elements and in terms of an intuitive picture of the increasing spatial extension of the ionized orbital along the dissociative-ionization pathway.

This summary clarifies the substantial advances and the fundamental differences of the present work compared to Ref. [18]. These advances are clearly explained in the second paragraph of the introduction, which reads as follows (p. 3):

“In this work, we introduce two new ideas that allow us to directly access and quantify the effect of nuclear motion on molecular photoionization delays. First, we perform a direct comparison of delays associated with dissociative and non-dissociative ionization to the same final state of the molecular cation. This eliminates the electronic effects caused by ionization from different orbitals and isolates the effect of nuclear motion in the measurement. Second, we also introduce the direct comparison of delays from different isotopologues (CH₄ vs. CD₄) to quantify the effect of the nuclear mass on the photoionization delays. Since the photoelectron leaves the parent ion within a few tens of attoseconds at typical photon energies in the extreme ultraviolet (XUV), only the fastest nuclear motions can be expected to leave an observable imprint on the photoionization delays.”

We further highlighted the advances of our work compared to Ref. [18] as follows (p. 8):

“It is indeed worth recalling that some effects attributed to nuclear motion have been found in the region of the Q1 resonances of H₂ [18] and the $3\sigma_g$ shape resonance of N₂ [10]. In the case of H₂ the autoionization lifetimes of the Q1 resonances are on the order of femtoseconds, which provides a long time interval for electron-nuclear coupling to manifest itself. In contrast, in the present study of methane, there is no visible trace of either shape or autoionizing Feshbach resonances in the investigated range of photon energies, such that the present work quantifies nuclear-motion effects in the general case of non-resonant photoionization for the first time.”

2. In their reply, the authors stated that "Finally, let us point out that the angle dependence of the photoionization delays is not the topic of this work". As shown in ref 18, measuring the photoelectron spectra in an angle-resolved way can provide a better analysis framework and a more comprehensive view of the dynamics. Therefore, "the angle dependence of the photoionization delays" may not be the topic of this work, but "the angle-resolved measurement of the photoelectron spectra" should be a significant part of the measurement to support the conclusion.

Based on elementary physical considerations of nuclear dynamics during a non-resonant (attosecond) photoionization process, one does not expect an angle dependence to those nuclear-motion effects. In the non-resonant photoionization studied in our work, the photoelectron wave packet leaves the molecule within tens of attoseconds or less. During this time interval, the nuclei have no time to move enough to noticeably distort the molecule. Therefore, we do not expect an angle dependence to the nuclear-motion effects.

This basic, physically motivated expectation is fully confirmed by angle-dependent photoionization delays, which we have now extracted from our experimental data and presented in Fig. S4. A corresponding sentence has been added to the main text (p. 7): “For completeness, we have also determined the angle-resolved photoionization delays (Fig. S4), but have found no angular dependence within the error bars of our measurements.”

3. In their reply, the authors stated that "We do indeed not expect any significant angle-dependence on the nuclear-motion effects studied in this work. Therefore, there was no need and no point in measuring (or calculating) the photoionization delays as a function of emission angle." I find this statement somewhat disappointing. If the authors wanted to show the "rule" as they indicated in the reply, they should perform a convincing study to confirm the "no need and no point" from their expectation (which is not so obvious to me) rather than throwing away doubts and suggestions. In sum, while the authors have addressed some of my concerns, I still do not see clear significance and novelty in the current work. Therefore, my original decision remains unaltered.

We refer to the reply to point 2 above. Our expectations that there was "no need and no point" to measure the angle-dependence of the nuclear-motion effects in the present work has been confirmed by the results that are now shown in Fig. S4, reproduced for the convenience of reviewer #2 as Fig. R1 below.

Fig. R1 Angle-resolved photoemission time delays: (a, b) Experimentally measured angle-resolved photoemission time delay differences of SB12 between (a) CH_3^+/H and CH_4^+ , (b) CD_3^+/D and CD_4^+ . The emission angle is defined with respect to the polarization axis of the XUV-APT. (c, d) Same as (a, b) but for SB14.

Now, let us discuss the feasibility of angle-resolved measurements in the molecular frame. First, we should note that Ref. 18 is about H_2 , where almost 5% of ionization goes to the dissociative channel and there is cylindrical symmetry, which reduces the information on angular orientation to two angles. In the present work, we are dealing with CH_4 and CD_4 , where the contribution of the dissociative channel is smaller and one needs 3 angles to access information about molecular orientation. Therefore, an experiment as that of Ref. 18 in CH_4 is nearly impossible. Molecular-frame measurements can be done in polyatomic molecules in some cases (e.g. in CF_4 , see Ref. 11). However, in this case, the dissociative channel is 100% of the total ionization probability and not all the angles defining the molecular orientation were determined, which made the experiment feasible.

These considerations change nothing to the fact that we do still not expect any angle-dependence (also not in the molecular frame) to the nuclear-motion effects in non-resonant (attosecond) photoionization, for the same reasons, as discussed above. In conclusion, our work has exhausted the current capabilities of experiment and theory alike on the subject of our investigation, i.e. nuclear-motion effects on the general case of attosecond photoionization. This work is the first of its kind, it reaches an unprecedented accuracy in the field of molecular photoionization delays and introduces new measurement concepts, as explained in response to point 1. We are therefore convinced that it has its place in *Nature Communications*.

REVIEWERS' COMMENTS

Reviewer #2 (Remarks to the Author):

The current version of the manuscript has not yet convinced me that it is suitable for publication in Nature Communications. Below are my comments.

1. Resonant and non-resonant interactions are just two equally important processes (and actually the resonant processes attract a lot more attention). Even the authors insist on putting their measurement of a non-resonant channel on a more general and important place, I personally do not agree with them. Their statement "non-resonant ionization is more general than near-resonant ionization because resonances are molecule-specific" is biased because non-resonant ionization is also target-specific and that's why photoionization delay can be measured to be different for different targets. The current work of showing a measurable photoionization delay in a faster dissociative channel does not provide fundamentally new information either technically or scientifically given the work of ref [18]. So my concerns that no clear significance and novelty of the current work is found is still poorly addressed.

2. I still believe that the angle-resolved measurement of the photoelectron spectra is a significant part of the study to support the conclusion, as successfully demonstrated in ref [18]. The authors response that "Based on elementary physical considerations of nuclear dynamics during a non-resonant (attosecond) photoionization process, one does not expect an angle dependence to those nuclear-motion effects". Although the "elementary physical considerations" seems intuitive, whether it is correct depends on the precision of the measurement. For example, the "photoemission" process was initially considered to take place instantly with no delay, the advancement of attosecond science makes it now possible to measure this very tiny time delay. Therefore, a measurement is definitely more reliable than the "elementary physical considerations" and thus is necessary.

The authors then claim that "This basic, physically motivated expectation is fully confirmed by angle-dependent photoionization delays, which we have now extracted from our experimental data and presented in Fig. S4", I appreciate the authors' effort to perform a more comprehensive data analysis. However, my understanding is that Fig. S4(Fig. R1) shows the lab-frame angular-resolved photoemission delay rather than the molecular-frame. Without a comparison with the theoretical treatment and interpretation, I do not know what this data means. My guess is that the lab-frame angular-resolved photoemission delay is the only viable measurement since they indicate that "an experiment as that of Ref. 18 in CH4 is nearly impossible". From this reply, my impression is twofold: 1) the technique demonstrated in this manuscript is not advanced as compared to previous one. 2) the conclusion is supported by a spatially averaged measurement that is less convinced than previously reported method.

3. This is probably a more serious concern. Based on the results in Fig. R1, it is difficult to draw the conclusion "we do not expect an angle dependence to the nuclear-motion effects". The variation depth reaches about 50%, and you probably can fit it with any kind of curves. The data only indicates that with the accuracy of the current setup, measuring a reliable angle-resolved photoemission delay is just out of reach. So the statement in their reply "Our expectations that there was "no need and no point" to

measure the angle-dependence of the nuclear-motion effects in the present work has been confirmed by the results that are now shown in Fig. S4" is not objective or rigorous.

In sum, I do not recommend its publication.

Reviewer #4 (Remarks to the Author):

No comments for author.

Reviewer #2 (Remarks to the Author):

The current version of the manuscript has not yet convinced me that it is suitable for publication in Nature Communications. Below are my comments.

1. Resonant and non-resonant interactions are just two equally important processes (and actually the resonant processes attract a lot more attention). Even the authors insist on putting their measurement of a non-resonant channel on a more general and important place, I personally do not agree with them. Their statement "non-resonant ionization is more general than near-resonant ionization because resonances are molecule-specific" is biased because non-resonant ionization is also target-specific and that's why photoionization delay can be measured to be different for different targets. The current work of showing a measurable photoionization delay in a faster dissociative channel does not provide fundamentally new information either technically or scientifically given the work of ref [18]. So my concerns that no clear significance and novelty of the current work is found is still poorly addressed.

We refer to our previous replies to similar comments by the same reviewer.

2. I still believe that the angle-resolved measurement of the photoelectron spectra is a significant part of the study to support the conclusion, as successfully demonstrated in ref [18]. The authors response that "Based on elementary physical considerations of nuclear dynamics during a non-resonant (attosecond) photoionization process, one does not expect an angle dependence to those nuclear-motion effects". Although the "elementary physical considerations" seems intuitive, whether it is correct depends on the precision of the measurement. For example, the "photoemission" process was initially considered to take place instantly with no delay, the advancement of attosecond science makes it now possible to measure this very tiny time delay. Therefore, a measurement is definitely more reliable than the "elementary physical considerations" and thus is necessary.

The authors then claim that "This basic, physically motivated expectation is fully confirmed by angle-dependent photoionization delays, which we have now extracted from our experimental data and presented in Fig. S4", I appreciate the authors' effort to perform a more comprehensive data analysis. However, my understanding is that Fig. S4(Fig. R1) shows the lab-frame angular-resolved photoemission delay rather than the molecular-frame. Without a comparison with the theoretical treatment and interpretation, I do not know what this data means. My guess is that the lab-frame angular-resolved photoemission delay is the only viable measurement since they indicate that "an experiment as that of Ref. 18 in CH₄ is nearly impossible". From this reply, my impression is twofold: 1) the technique demonstrated in this manuscript is not advanced as compared to previous one. 2) the conclusion is supported by a spatially averaged measurement that is less convinced than previously reported method.

We refer to our previous replies to similar comments by the same reviewer.

3. This is probably a more serious concern. Based on the results in Fig. R1, it is difficult to draw the conclusion "we do not expect an angle dependence to the nuclear-motion effects". The variation depth reaches about 50%, and you probably can fit it with any kind of curves. The data only indicates that with the accuracy of the current setup, measuring a reliable angle-resolved photoemission delay is just out of reach. So the statement in their reply "Our expectations that there was "no need and no point" to measure the angle-dependence of the nuclear-motion effects in the present work has been confirmed by the results that are now shown in Fig. S4" is not objective or rigorous. In sum, I do not recommend its publication.

We refer to our previous replies to similar comments by the same reviewer.

Reviewer #4 (Remarks to the Author):

No comments for author.

We thank reviewer 4 for their time spent on reviewing our manuscript.